# EXPLOITING NATURAL FREQUENCY DEVIATION FOR DIFFUSION-GENERATED IMAGE DETECTION

## ABSTRACT

Diffusion models have achieved remarkable success in image synthesis, but the generated high-quality images raise concerns about potential malicious use. Existing detectors often struggle to capture distinctive features across different training models, limiting their generalization to unseen diffusion models with varying schedulers and hyperparameters. To address this issue, we observe that diffusion-generated images exhibit progressively larger differences from real images across low- to high-frequency bands. Based on this insight, we propose a novel image representation called **N**atural **Fre**quency **De**viation (**DEFEND**). DEFEND applies a weighted filter to the Fourier spectrum, suppressing less discriminative bands while enhancing more informative ones. This approach, grounded in a comprehensive analysis of frequency-based differences between real and diffusion-generated images, enables robust detection of images from unseen diffusion models and provides resilience to various perturbations. Extensive experiments on diffusion-generated image datasets show that our method outperforms state-of-the-art detectors with superior generalization and robustness.

## 1 INTRODUCTION

Diffusion models (Ho et al., 2020; Dhariwal & Nichol, 2021) have achieved remarkable success in image synthesis, producing high-quality and diverse results. However, the easy accessibility and realistic generated images present a significant challenge, as they can be easily misused for malicious purposes, such as fabricating evidence or misleading the public, raising serious social, privacy, and ethical concerns (Devlin & Cheetham, 2023). Therefore, how to detect diffusion-generated images has become an urgent and critical issue recently.

Recent methods for detecting diffusion-generated images focus on specific model characteristics, such as using reconstruction error (Wang et al., 2023), leveraging pre-trained vision-language models (Ojha et al., 2023), or identifying artifacts introduced by the upsampling layers (Tan et al., 2024). However, these approaches are limited in their generalization to different diffusion models. They often rely on specific models for reconstruction, require relevant generated images as reference sets, or depend on architectural features like upsampling layers, making them less effective when detecting images from unknown or unseen diffusion models.

With all the above concerns in mind, we raise the following question: Can we develop a general diffusion-generated image detector based on their inherent difference with natural real images? As we do not rely on specific diffusion-generated images, the detector should be sufficiently general and robust. To this end, we first analyze the difference between natural real images and diffusion-generated images in the frequency domain, as the frequency domain contains more distinguishable information than the pixel domain (Van der Schaaf & van Hateren, 1996), as shown in Fig. 1. We can observe that there exists a clear discrepancy between natural real images and diffusion-generated images in their Fourier spectrum, specifically in the mid- and high-frequency band, which could serve as discriminative clues for detection.

To leverage this, we first conduct a comprehensive frequency analysis on diffusion-generated images and natural real images to explore the intrinsic discrepancy clues, which indicates that the discriminability increases from low- to high-frequency bands. Based on our analysis, we further propose a general image representation termed **N**atural **Fre**quency **De**viation (**DEFEND**): By designing a frequency-selective function that serves as the weighted filter banks, it restrains the less discriminative

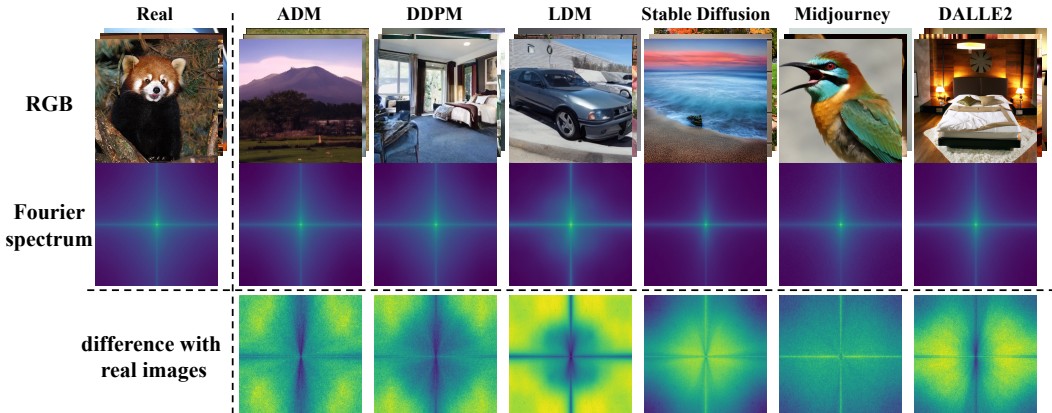

Figure 1: **The magnitude difference between real image and different diffusion-generated models.** The fake images generated by different diffusion models (top) leave traces in their Fourier spectrum (middle). We explore their differences with natural real images for the detection of the diffusion-generated images (bottom). The Fourier spectrums are averaged on 1000 sampled images. The darker the color, the smaller the magnitude; the lighter the color, the larger the magnitude.

bands (*i.e.*, low frequency) and enhances more significant discriminative frequency bands (*i.e.*, high-frequency) in the Fourier spectrum, thus leading to more discriminative representation. Compared to detectors only focusing on certain bands, *i.e.*, high frequency, our representation can exploit the clues existing in all different bands, which should be more general and robust to different diffusion models and various perturbations. Extensive experiments on various public diffusion-generated image datasets demonstrate the superiority of our proposed method against other state-of-the-art competitors. Our main contributions are summarized as follows:

- We conduct a comprehensive frequency analysis on natural real images and diffusion-generated fake images. We find that the diffusion-generated images exhibit increasingly significant differences with natural real images, from low- to high-frequency bands.

- To leverage this, we propose **DEFEND** as a more discriminative representation for detection, by designing a frequency-selective function that serves as weighted filter banks for restraining the less discriminative bands (*i.e.*, low-frequency) and enhancing the more discriminative frequency bands(*i.e.*, high-frequency) in the Fourier spectrum.

- Extensive experiments on various public datasets demonstrate the superiority of our proposed method against other state-of-the-art competitors in detecting diffusion-generated images with impressive generalization and robustness.

## 2    RELATED WORK

**Diffusion models.** Diffusion Models have achieved remarkable success in image synthesis task (Ho et al., 2020; Dhariwal & Nichol, 2021; Rombach et al., 2022). The main idea of diffusion models is inspired by the non-equilibrium thermodynamics proposed in (Sohl-Dickstein et al., 2015). Typically, diffusion models define two Markov chains of diffusion steps that first slowly add Gaussian noise to clean images, until disturbing them into isotropic Gaussian noise (termed diffusion or forward process); then they learn to reverse the diffusion process to generate clean samples from the noise (termed denoising or reverse process). Due to substantial efforts focusing on improving model architectures (Rombach et al., 2022), sampling methods (Song et al., 2020; Lu et al., 2022), and optimizing processes (Ho & Salimans, 2022; Nichol & Dhariwal, 2021), recent diffusion models are capable of generating high-quality images beyond human imagination at an extremely low cost, which can be a double-edged sword. Thus, developing general and robust diffusion-generated image detectors has recently become a critical issue.

**Generated image detection.** Recent generative models, such as GANs (Goodfellow et al., 2014; Karras et al., 2018; 2019; Brock et al., 2018) and Diffusion models (Dhariwal & Nichol, 2021;

Nichol et al., 2021; Rombach et al., 2022; Ramesh et al., 2022), have achieved remarkable success in image generation task. Various detectors have been proposed to prevent the malicious use of generated images (Chai et al., 2020; Qian et al., 2020). To develop a general GAN-generated image detector,(Wang et al., 2020) introduce carefully designed pre- and post-processing with data augmentation. To detect generated and manipulated images,(Chai et al., 2020) propose to use patch-level artifacts. Recently, (Wang et al., 2023) found that diffusion-generated images are easier to reconstruct by diffusion models than real images. They propose a representation DIRE based on reconstruction error. (Ojha et al., 2023) propose to use the pre-trained vision-language models to learn discriminative clues for detection. NPR (Tan et al., 2024) explore the artifacts introduced by the up-sampling layer in diffusion model architectures. These methods still, however, highly rely on specific patterns in training diffusion-generated images, which could lead to performance drops when detecting unseen models. Instead, we exploit the intrinsic statistic difference with natural real images in the frequency domain to discriminate from diffusion-generated images.

**Frequency artifacts in generated images.** Some prior works have demonstrated that generated images exist artifacts in the frequency domain (Dzanic et al., 2020; Ricker et al., 2022; Corvi et al., 2023a; Yu et al., 2019; Chandrasegaran et al., 2021). To detect GAN-generated images,(Frank et al., 2020) analyzes artifacts by discrete cosine transform (DCT). (Dzanic et al., 2020) leverage the Fourier spectrum discrepancy in GAN- and VAE-generated images. (Qian et al., 2020) propose exploring the frequency clues to detect generated and manipulated images. There are already some works that find specific patterns in diffusion-generated images in the frequency domain which could serve as clues for detection, such as (Ricker et al., 2022) analyze the frequency fingerprints of different diffusion models, (Corvi et al., 2023a;b) analyze the artifacts in both spatial and frequency domains, (Li et al., 2024) train a mask on Fourier spectrum and use the cosine similarity with a reference set for detection. These methods, however, focus mainly on the frequency distribution of specific diffusion-generated images, which ignores the natural real images' inherent distributions and may lead to limited generalization. In this paper, we instead focus on the general frequency difference between natural real images and diffusion-generated images and propose a new image representation by restraining the less discriminative bands and enhancing the more discriminative ones.

## 3 METHODOLOGY

### 3.1 FREQUENCY ANALYSIS ON NATURAL REAL AND DIFFUSION-GENERATED IMAGES

We first analyze the difference between natural real images and diffusion-generated images in the frequency domain, as shown in Fig 3 (a). Specifically, to transform an image $\mathbf{x}$ from the spatial domain to the frequency domain, we first transform it into a grayscale image since the color information contributes less to the frequency distribution of an image. Then we compute the Discrete Fourier Transform and the mean power spectrum $F(\mathbf{x})$ that can be formulated as follows:

$$F(\mathbf{x}) = \log|\text{DFT}(\mathbf{x})|^2, \tag{1}$$

where $\mathbf{x}$ is the input image, the $\text{DFT}(\cdot)$ is the Discrete Fourier Transform and $|\cdot|$ computes the magnitude on each pixel. In practice, we use the FFT algorithm to compute the DFT. We compute and visualize the mean power spectrum on images generated from different diffusion models and different time steps, as shown in Fig. 2 (a) and (b), respectively.

From the results, we observe that the diffusion-generated images and natural real images exhibit significant differences in the frequency domain: their discrepancy becomes increasingly discriminative from low- to high-frequency bands. This phenomenon can be reflected in pixel space as the diffusion-generated images are usually smoother and lack the high-fidelity details that indicate the high-frequency parts. Besides, the images generated by different diffusion models and different timesteps exhibit different frequency patterns, and more time steps lead to high similarity to real images. And all of the generated images follow the above observation that their discrepancies with real images become increasingly discriminative from the low to high frequencies. Thus, we can draw following conclusion from above analysis:

**Remark 1: Diffusion-generated images exhibit significant discrepancies with natural real images. These discrepancies are increasingly discriminative from low- to high-frequency bands.**

Furthermore, we investigate what causes this phenomenon during the diffusion forward/backward processes. Specifically, we analyze the mean power spectrum of the intermediate results of the

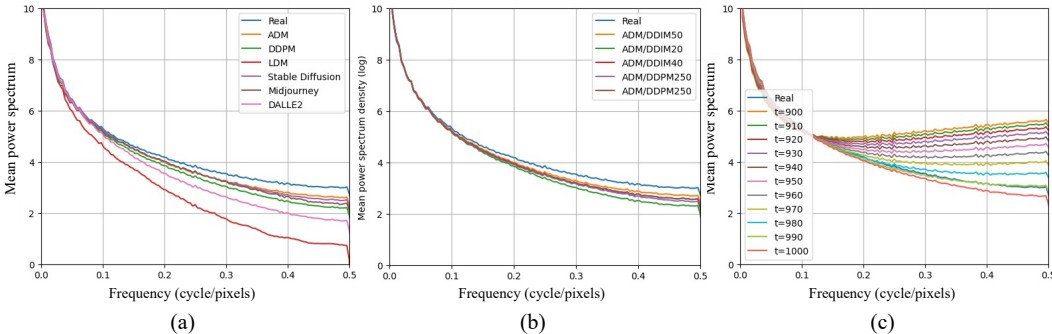

Figure 2: **Mean power spectrum** of natural real and diffusion-generated images from different diffusion models (a) and different time steps (b). We further explore the spectrum during the denoising process in (c).

last 100 time steps during the DDPM (Ho et al., 2020) 1000-step denoising process by using the ADM (Dhariwal & Nichol, 2021), as illustrated in Fig. 2 (c). We observe that the mid-high-frequency parts of generated images are generated towards mainly the end steps of the denoising process. And these end steps determine the underestimation of mid-high-frequency bands. Note that during training of diffusion models, a neural network $\epsilon_\theta$ is optimized to predict the added noise, given the noisy image $\mathbf{x}_t$ and corresponding time step $t$, the optimization target is a sampling and denoising process which can be defined as follows:

$$L_\theta(\mathbf{x}_0, t) = \|\epsilon - \epsilon_\theta(\sqrt{\bar{\alpha}_t}\mathbf{x}_0 + \sqrt{1 - \bar{\alpha}_t}\epsilon, t)\|^2, \tag{2}$$

where $\epsilon \sim \mathcal{N}(\mathbf{0}, \mathbf{I})$, $\mathbf{x}_0$ is clean image, and $\alpha_t$ is the predefined noise schedule. By analyzing the optimization process above, we argue that the underestimation at the end steps is caused by the optimization objective described in Eq. 2 because the generated images towards the end steps are closer to denoised clean images. This makes it more difficult to predict the added noise when compared to pure Gaussian distributions, but the Eq. 2 treats equally the denoising tasks at different noise levels. There are also some existing analyses (Nichol & Dhariwal, 2021; Ricker et al., 2022) that agree that this objective cannot lead to good likelihood values. Thus, we can draw following conclusion for the cause of the frequency discrepancy:

**Remark 2: The spectrum discrepancy is highly related to the challenging optimization objective, when towards denoising step $t = 0$.**

Moreover, some prior studies (Van der Schaaf & van Hateren, 1996; Field, 1987; Burton & Moorhead, 1987) show that the mean power spectrum of natural real images has the following rule:

$$S(f) \propto f^{-\alpha}, \alpha \approx 2, \tag{3}$$

where $S(\cdot)$ is the mean power spectrum and $f$ is the frequency.

### 3.2 NATURAL FREQUENCY DEVIATION REPRESENTATION

With the above observation that diffusion-generated images exist frequency discrepancy with natural real images, it comes to our mind that if we could design a representation that exploits the most discriminative clues and removes the similar patterns in the frequency domain, we could obtain a more effective representation for distinguishing the diffusion-generated images from real ones. To this end, we propose a novel image representation, termed **N**atural **Fr**equency **De**viation (**DEFEND**) for diffusion-generated image detection: it restrains the less discriminative (low frequency) bands and enhances those more discriminative (mid-high frequency), as shown in Fig 3 (b). As the representation is designed by the general observation and the principle on natural real and diffusion-generated fake images, our representation should be general and robust for the detection task.

To achieve this, we first compute the frequency spectrum deviation between natural real and diffusion-generated images, based on Fig. 2 (a)&(b). Specifically, we compute the subtraction of each spectrum with the one on natural real images, and visualize them with a scaling factor, as shown in Fig. 4 (a)&(b). We aim to design a frequency-selective function $w(\cdot)$ based on the distribution above to

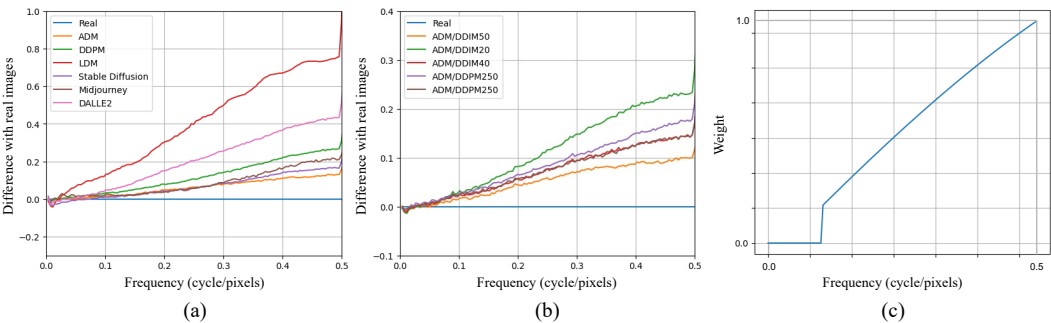

Figure 3: **Overview of our proposed method.** We first analyze the discrepancy of mean power spectrum between natural real and diffusion-generated images, as shown at the top (a). Based on the analysis, we design a specific weight function $w(f)$ that serves as the filter banks on the Fourier spectrum to restrain the less discriminative frequency bands and to enhance the more discriminative ones, thus leading to more discriminative representation, as shown at the bottom (b).

serve as the weighted filter banks applying on the Fourier spectrum to restrain the less discriminative bands and to enhance the more discriminative bands. Then, we inverse the enhanced Fourier spectrum to RGB space, thus leading to a more discriminative representation than the original RGB images. This process can be formulated as follows:

$$\text{DEFEND}(\mathbf{x}) = \text{IDFT}(\text{DFT}(\mathbf{x}) \cdot w(f)), \tag{4}$$

where the $\mathbf{x}$ is the input image, $\text{DFT}(\cdot)$ is the Discrete Fourier Transform, and the $\text{IDFT}(\cdot)$ is the Inverse Discrete Fourier Transform. In practice, we use the FFT algorithm to compute them.

Figure 4: **Spectrum discrepancy** between natural real and diffusion-generated images in (a) and (b). We further design the **weight function** based on the discrepancy as shown in (c).

For the frequency-selective function $w(\cdot)$, we introduce two following principles based on the analysis above to process the less discriminative band (*i.e.*, low frequency), and more discriminative band (*i.e.*, mid-high frequency), respectively, described as follows:

**Low-frequency band.** In Fig. 4 (a)&(b), the low-frequency part of real and diffusion-generated images exhibits high similarity, which indicates that there is no significant discrepancy in this band. Hence, we remove the low-frequency information to restrain the less discriminative band by simply setting the weight to zero, formulated as follows:

$$w(f) = 0, f \le \tau, \tag{5}$$

where $\tau$ is the threshold for the low frequency, and we empirically set $\tau = 0.1$.

**Mid-high frequency band.** The mid-high frequency parts are increasingly discriminative, as indicated in Fig. 4 (a)&(b). Following the principle that higher weights should be assigned to more discriminative bands, we compute the weights, based on their discrepancy. To this end, we introduce another kernel function $k(\cdot)$ to fit the power spectrum discrepancy distribution in Fig. 4 (a)&(b), which can be formulated as follows:

$$k(f) = |log|G_1(f)|^2 - log|G_0(f)|^2|, \tag{6}$$

where $\{G_1(f), G_0(f)\}$ are the Discrete Fourier Transform distribution of diffusion-generated and natural real images, respectively. As we only care about the discrepancy between them, we can simplify the above equation as follows:

$$|G_1(f)| = e^{\frac{k(f)}{2}} \cdot |G_0(f)|. \tag{7}$$

Note that, for natural real images, their frequency distribution follows the principle described in Eq. 3. Therefore, we choose $\alpha = 2$ which should be an appropriate parameter to approximate the statics of real images, formulated as:

$$S_0(f) = |G_0(f)|^2 = \frac{1}{f^2}. \tag{8}$$

Thus, we can further compute the discrepancy of each frequency band, based on Eq. 7 & 8 to obtain the desired weight function as follows:

$$w(f) = ||G_1(f)| - |G_0(f)|| = (e^{\frac{k(f)}{2}} - 1) \cdot \frac{1}{f} \tag{9}$$

Considering both the low and mid-high frequency described above, our final designed frequency-selective weight function is as follows:

$$w(f) = \begin{cases} 0, & f \leq \tau \\ (e^{\frac{k(f)}{2}} - 1) \cdot \frac{1}{f}, & f > \tau \end{cases} \tag{10}$$

Furthermore, based on our observations, we empirically choose the two-degree linear function (quadratic function) as the kernel function with the coefficients $k(f) = -0.2f^2 + 0.8f - 0.05$ by fitting the distributions. The corresponding designed function $w(f)$ is shown in Fig. 4 (c), which restrains the less discriminative band, (low frequency), and enhances the more discriminative band (mid-high frequency), thus leading to a more discriminative representation.

## 3.3 DIFFUSION-GENERATED IMAGE DETECTION

After the representation learning stage, we can obtain the DEFEND representations for both natural real and diffusion-generated images. We further use the representations as input to train a naive binary classifier to distinguish the real and generated images by a simple binary cross-entropy loss, which is formulated as follows:

$$L(y, \hat{y}) = -\Sigma_{i=1}^{n} (y_i \log(\hat{y}_i) + (1 - y_i) \log(1 - \hat{y}_i)), \tag{11}$$

where $n$ is the mini-batch size, $y \in \{0, 1\}$ is the ground-truth label for real and fake, and $\hat{y}$ is the output prediction of the classifier. We choose ResNet-50 (He et al., 2016) with a fully-connected layer as our classifier. And during the inference stage, we input the DEFEND representation to the trained classifier that could be classified as real or diffusion-generated.

# 4 EXPERIMENT

## 4.1 EXPERIMENTAL SETUP

**Dataset.** Following recent state-of-the-art diffusion-generated image detectors (Wang et al., 2023; Ojha et al., 2023; Tan et al., 2024; Zhu et al., 2024), we evaluate our proposed method on three public diffusion-generated image datasets, including (1) GenImage (Zhu et al., 2024), (2) UniformerDiffusion (Ojha et al., 2023), and (3) DiffusionForensics (Wang et al., 2023), described in detail below. (1) The GenImage dataset is a recent challenging dataset containing seven different diffusion models trained on ImageNet, with a broad range of image classes, including ADM (Dhariwal & Nichol, 2021), Glide (Nichol et al., 2021), Midjourney (Midjourney, 2023), Stable-Diffusion-v1.4(Rombach et al., 2022), Stable-Diffusion-v1.5(Rombach et al., 2022), VQDM (Gu et al., 2022), Wukong (Wukong, 2022). (2) The UniformerDiffusion dataset contains images generated from different diffusion models with various settings, such as different timesteps. It includes ADM, LDM (Rombach et al., 2022), Glide, and DALLE (Ramesh et al., 2021). (3) The DiffusionForensics dataset contains various different recent diffusion models on LSUN-Bedroom dataset, including ADM, DDPM (Ho et al.,

2020), iDDPM (Nichol & Dhariwal, 2021), PNDM (Liu et al., 2022), Stable-Diffusion-v1, Stable-Diffusion-v2, LDM, VQDM, IF (Saharia et al., 2022), DALLE2 (Ramesh et al., 2022), Midjourney. For training set, we use the fake images generated from ADM trained on ImageNet and real images from ImageNet, which contain 40,000 fake and real images, respectively.

**Evaluation metric.** Following prior state-of-the-art methods (Wang et al., 2020; 2023; Ojha et al., 2023), we report the average precision (AP) and accuracy (ACC) with a fixed 0.5 threshold.

**Baselines.** For fair and comprehensive comparisons, we choose and categorize four different types of state-of-the-art detectors: traditional image classification backbones (including (1) ResNet-50 (He et al., 2016) and (2) Swin-T (Liu et al., 2021)), deepfake detectors (including (3) Patchfor (Chai et al., 2020) and (4) F3Net (Qian et al., 2020)), diffusion-generated image detectors ((5) DIRE (Wang et al., 2023)), and universal detectors (including (6) CNNDet (Wang et al., 2020), (7) uniFD (Ojha et al., 2023) and (8) NPR (Tan et al., 2024)). We train all aforementioned baselines by using the same training set, from scratch with their released code. Please refer to the appendix for more details.

## 4.2 COMPARISON TO THE STATE-OF-THE-ART

**Generalization to unknown models.** We first evaluate the generalization of our proposed on unknown diffusion models, which is a major challenge in this task. Specifically, we train all detectors with the same training dataset generated from ADM on ImageNet, then we evaluate them on the three aforementioned public datasets. We first evaluate on the challenging GenImage, which is a recent and diversified dataset with multi classes trained on ImageNet. The ACC/AP results are presented in Tab. 1. From the results, we observe that all baseline detectors have a slight performance drop when encountering more diversified generated images, which is a challenging setting for existing detectors. Among these detectors, our method still achieves impressive generalization with 99.91% average ACC, with 5.41% and 10.98% AP improvements compared to the recent DIRE and NPR.

Furthermore, we evaluate the UniformerDiffusion dataset that contains various settings, such as different time steps. The results are shown in Tab. 2. Our method achieves 5.63% and 5.04% AP improvements compared to recent DIRE and NPR. We also observe that naive detectors, such as ResNet-50 and Swin-T, cannot achieve desired performance on diffusion-generated images. Other detectors designed for GAN-generated, forgery, or universal fake images could all achieve competitive performance, yet they still suffer performance drops, when detecting specific unknown diffusion models, such as CNNDet and UniFD on DALLE. Our method maintains the same impressive performance on all different diffusion models and with various settings. This provides support for the impressive generalization of our method to various settings of diffusion models.

Moreover, we evaluate on the DiffusionForensics dataset that contains more unknown diffusion models, *i.e.*, 11 different diffusion models, including recent DALLE-2 and Midjourney. The ACC/AP results are shown in Tab. 3. Our proposed method also achieves impressive ACC and AP across more recent and different diffusion models, such as Stable-Diffusion, DALLE-2, and Midjourney, with 10.12% and 3.79% AP improvements compared to CNNDet and UniFD. This indicates the potential of our method for detecting future challenging diffusion models.

Table 1: **Generalization results on GenImage dataset.** We report the detection accuracy and average precision (ACC/AP) averaged over real and fake images on unknown diffusion models.

| Detection method | Different Diffusion Models in GenImage | | | | | | | Total |
|---|---|---|---|---|---|---|---|---|
| | ADM | Glide | Midjourney | SD-v1.4 | SD-v1.5 | VQDM | Wukong | Avg. |
| ResNet-50 | 81.20/97.42 | 80.01/93.05 | 60.85/66.55 | 55.45/60.71 | 54.10/60.42 | 76.40/87.73 | 51.01/53.59 | 65.57/74.21 |
| Swin-T | 74.84/88.45 | 75.69/93.59 | 62.48/70.85 | 70.69/78.65 | 71.19/78.02 | 73.49/75.61 | 70.03/77.46 | 71.20/80.38 |
| Patchfor | 99.65/99.43 | 99.81/99.63 | 57.19/85.88 | 50.59/61.62 | 50.64/61.51 | 99.83/99.95 | 50.52/61.89 | 72.60/81.42 |
| F3Net | 99.64/99.99 | 99.84/99.99 | 51.48/74.93 | 50.02/59.41 | 50.33/61.98 | 99.94/99.99 | 50.13/52.61 | 71.63/78.41 |
| DIRE | 61.35/97.91 | 61.65/99.17 | 61.65/94.83 | 59.55/92.09 | 59.30/92.94 | 61.05/96.88 | 58.70/88.31 | 60.46/94.59 |
| CNNDet | 64.55/83.73 | 62.45/70.72 | 51.15/50.69 | 56.30/54.72 | 54.30/54.08 | 62.70/69.97 | 56.85/57.76 | 58.33/63.10 |
| UniFD | 72.45/91.45 | 62.30/63.65 | 53.50/50.83 | 67.00/78.58 | 67.10/74.38 | 72.25/95.35 | 70.45/85.94 | 66.44/77.17 |
| NPR | 77.90/96.91 | 77.95/93.69 | 73.30/86.76 | 75.40/83.14 | 73.50/83.40 | 80.15/91.35 | 74.00/87.88 | 76.03/89.02 |
| **DEFEND** | **99.95/100.0** | **99.95/100.0** | **99.95/100.0** | **99.90/99.99** | **99.95/100.0** | **99.90/100.0** | **99.80/100.0** | **99.91/100.0** |

Table 2: **Generalization results on UniformerDiffusion dataset.** We report the detection accuracy and average precision (ACC/AP) averaged over real and fake images on unknown diffusion models.

| Detection method | Different Diffusion Models in UniformerDiffusion | | | | | | | | Total |
| | ADM | LDM | | | Glide | | | DALLE | Avg. |
| | | 200 steps | 200 w/ CFG | 100 steps | 100 & 27 | 50 & 27 | 100 & 10 | | |
|---|---|---|---|---|---|---|---|---|---|
| ResNet-50 | 81.95/97.95 | 78.40/89.24 | 74.70/86.17 | 78.60/89.92 | 80.35/94.36 | 80.95/95.47 | 81.10/95.29 | 76.70/89.52 | 79.09/92.24 |
| Swin-T | 79.74/84.82 | 78.94/80.08 | 75.44/80.01 | 78.19/79.60 | 79.89/91.82 | 79.98/90.96 | 80.04/92.40 | 77.24/84.35 | 76.68/85.51 |
| Patchfor | 99.71/99.91 | 99.69/99.98 | 99.61/99.97 | 99.69/**100.0** | 99.56/99.97 | 99.58/99.97 | 99.55/99.97 | 99.54/99.97 | 99.62/99.97 |
| F3Net | 99.49/**100.0** | 99.54/99.99 | 99.49/99.99 | 99.54/99.91 | 99.44/99.97 | 99.50/99.98 | 99.49/99.98 | 99.49/99.93 | 99.50/99.97 |
| DIRE | 62.05/98.68 | 59.85/90.97 | 58.00/86.30 | 60.40/91.71 | 62.00/98.92 | 62.05/**100.0** | 62.01/98.95 | 59.45/89.42 | 60.73/94.37 |
| CNNDet | 68.95/94.31 | 63.20/67.56 | 57.35/56.07 | 63.90/71.44 | 66.05/77.16 | 66.75/80.46 | 66.30/77.64 | 63.80/69.19 | 64.54/74.23 |
| UniFD | 73.30/93.35 | 73.20/95.94 | 64.45/75.67 | 73.35/95.83 | 70.80/82.33 | 70.50/83.24 | 69.90/82.13 | 70.75/87.32 | 70.78/86.98 |
| NPR | 83.15/96.32 | 83.25/92.34 | 83.10/93.69 | 83.25/93.37 | 81.90/95.86 | 82.45/96.51 | 82.60/96.52 | 83.05/95.10 | 82.84/94.96 |
| **DEFEND** | 100.0/100.0 | 99.90/100.0 | 100.0/100.0 | 100.0/100.0 | 100.0/100.0 | 100.0/100.0 | 100.0/100.0 | 100.0/100.0 | 99.99/100.0 |

Table 3: **Generalization results on DiffusionForensics dataset.** We report the detection accuracy and average precision (ACC/AP) averaged over real and fake images on unknown diffusion models.

| Detection method | Different Diffusion Models in DiffusionForensics | | | | | | | | | | | Total |
| | ADM | DDPM | iDDPM | PNDM | SD-v1 | SD-v2 | LDM | VQDM | IF | DALLE-2 | Midjourney | Avg. |
|---|---|---|---|---|---|---|---|---|---|---|---|---|
| ResNet-50 | 61.95/90.07 | 59.45/77.73 | 61.90/90.45 | 92.50/99.43 | 96.25/99.71 | 97.15/99.77 | 99.35/99.98 | 99.45/100.0 | 99.20/99.98 | 95.73/99.05 | 92.00/62.86 | 86.81/92.64 |
| Swin-T | 59.18/96.35 | 62.54/97.12 | 60.33/98.10 | 55.03/97.64 | 94.90/99.86 | 94.69/99.94 | 98.29/99.98 | 95.79/99.98 | 98.79/99.96 | 99.99/99.99 | 100.0/99.99 | 83.59/98.99 |
| Patchfor | 61.20/92.08 | 59.65/84.47 | 57.52/88.39 | 97.40/99.73 | 51.75/68.86 | 50.75/68.77 | 91.45/97.98 | 91.63/97.98 | 91.86/99.24 | 65.67/54.61 | 89.54/84.73 | 73.49/85.17 |
| F3Net | 90.24/98.35 | 94.96/98.83 | 94.39/99.01 | 98.44/99.86 | 51.03/53.87 | 51.08/61.56 | 98.94/99.99 | 98.94/99.99 | 98.94/99.99 | 65.44/71.09 | 89.08/94.62 | 84.68/88.83 |
| DIRE | 85.00/99.76 | 83.09/99.89 | 85.05/99.97 | 83.45/96.82 | 85.05/99.96 | 85.06/**100.0** | 85.01/99.98 | 85.05/99.97 | 85.00/99.97 | 80.07/99.94 | 72.82/99.89 | 83.15/99.65 |
| CNNDet | 70.75/82.84 | 66.91/81.64 | 72.35/86.42 | 72.60/92.97 | 72.65/89.18 | 70.25/78.86 | 71.55/87.75 | 72.95/**100.0** | 72.85/95.96 | 63.93/95.74 | 50.64/97.35 | 68.86/89.88 |
| UniFD | 66.15/91.11 | 79.50/96.82 | 87.30/98.31 | 91.25/98.83 | 93.05/99.42 | 85.25/98.22 | 55.25/88.49 | 95.00/99.71 | 58.35/87.76 | 99.50/**100.0** | 90.50/99.64 | 81.92/96.21 |
| NPR | 99.50/99.78 | 99.83/99.77 | 99.55/99.71 | 99.85/99.83 | 99.85/99.69 | 99.75/99.73 | 99.70/99.79 | 99.85/99.77 | 99.85/**100.0** | 99.80/99.53 | 99.73/96.31 | 99.75/99.45 |
| **DEFEND** | 100.0/100.0 | 100.0/100.0 | 100.0/100.0 | 100.0/100.0 | 100.0/100.0 | 100.0/100.0 | 100.0/100.0 | 100.0/100.0 | 100.0/100.0 | 100.0/100.0 | 100.0/100.0 | 100.0/100.0 |

The impressive performance across the three aforementioned datasets further demonstrates the superiority of our proposed DEFEND representation, as it restrains the less discriminative clues and enhances those more discriminative in the frequency domain for detection.

**Robustness to unseen perturbations.** The robustness to unseen perturbations is also a major concern for existing detectors, as there are various but common post-preprocessing perturbations in real-scenario applications, such as compression. To address this issue, we evaluate all detectors' robustness against three common but widely used perturbations on images generated from ADM (the same as the training set), including Gaussian Noise, Gaussian Blur, and JPEG Compression, following (Wang et al., 2020; 2023). For each perturbation, we employ three different severity levels to disrupt images: $\sigma = 0.001, 0.005, 0.01$ for Gaussian Noise, $\sigma = 1, 2, 3$ for Gaussian Blur, and $quality = 75, 50, 25$ for JPEG Compression. The results are shown in Fig. 5. From the results, we first observe that existing detectors would suffer from common perturbations, especially for Gaussian Noise. This indicates that some of the representations these detectors rely on might not be sufficiently robust to real scenario disruptions. Our proposed representation suffers significantly less from the above three perturbations, with only slight or even no performance drops. This indicates that, by exploring the discriminative clues with natural real images across all frequency bands, our proposed representation has impressive robustness against common perturbations.

## 4.3 ABLATION STUDY

**Comparison with different image representations.** To examine whether our proposed representation is better than other image representations for detecting diffusion-generated images, we first conduct further ablation studies on various inputs for detection, including RGB and grayscale images. The results on GenImage dataset are presented in Tab. 4, which indicates that RGB and grayscale images cannot achieve the desired generalization on unknown diffusion models. One explanation could be that pixel space does not share common distributions among different diffusion models. Their comparisons with our proposed DEFEND demonstrate that our representation serves as a general and robust image representation, thus contributing to a generalizable detector than simply using RGB images. This also provides more evidence for the superiority of our method by exploring the discriminative clues with natural real images in the frequency domain.

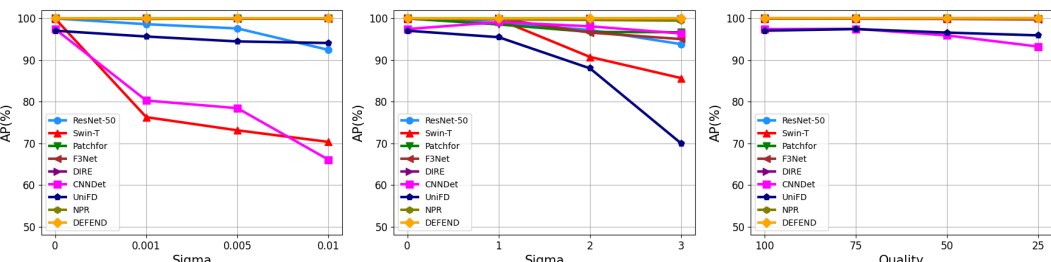

Figure 5: **Robustness results to unseen perturbations.** Average precision (AP) of different methods, when detecting real/fake images under three different types of perturbations with three different severity levels: Gaussian Noise ($\sigma = 0.001, 0.005, 0.01$), Gaussian Blur ($\sigma = 1, 2, 3$), and JPEG Compression ($quality = 75, 50, 25$) (from left to right).

Table 4: **Ablation study on different image representation.** We report the ACC/AP results on the GenImage dataset that indicates our designed representation can achieve improved performance.

| Representation | Different Diffusion Models in GenImage | | | | | | | Total |
|---|---|---|---|---|---|---|---|---|
| | ADM | Glide | Midjourney | SD-v1.4 | SD-v1.5 | VQDM | Wukong | Avg. |
| RGB | 78.40/96.28 | 76.95/89.49 | 60.10/65.20 | 55.60/60.94 | 53.75/59.68 | 71.65/80.70 | 50.60/54.01 | 63.86/72.33 |
| Grayscale | 97.90/99.75 | 98.00/99.81 | 75.85/89.44 | 65.65/81.87 | 65.65/82.68 | 92.90/98.61 | 67.00/83.52 | 80.42/90.81 |
| **DEFEND** | 99.95/100.0 | 99.95/100.0 | 99.95/100.0 | 99.90/99.99 | 99.95/100.0 | 99.90/100.0 | 99.80/100.0 | 99.91/100.0 |

**Effect of the minimum threshold on low frequency.** We conduct further ablation studies on the threshold for restraining low-frequency bands by employing a different minimum threshold $\tau$ or not, as presented in Tab. 5. We observe that the performance is improved when employing a suitable minimum threshold to restrain the low-frequency band. This demonstrates that low-frequency band cannot provide discriminative information for diffusion-generated image detection and that eliminating them could boost the performance. Additionally, the performance will drop when the threshold is too low or too high, which indicates that both introducing too much low-frequency information or ignoring too much mid-high frequency information could undermine the performance.

Table 5: **Ablation study on low-frequency bands.** We report the ACC/AP results on the GenImage dataset that indicates that introducing both too much low-frequency and ignoring too much mid-high-frequency information can undermine the performance.

| Minimum threshold $\tau$ | Different Diffusion Models in GenImage | | | | | | | Total |
|---|---|---|---|---|---|---|---|---|
| | ADM | Glide | Midjourney | SD-v1.4 | SD-v1.5 | VQDM | Wukong | Avg. |
| 0.00 | 99.35/99.99 | 99.80/99.99 | 99.75/99.95 | 99.30/99.96 | 99.20/99.98 | 99.65/99.96 | 98.75/99.96 | 99.40/99.97 |
| 0.05 | 99.90/100.0 | 99.85/100.0 | 99.95/100.0 | 99.80/99.99 | 99.85/100.0 | 99.90/100.0 | 99.75/100.0 | 99.86/100.0 |
| 0.20 | 99.75/100.0 | 99.85/100.0 | 99.95/100.0 | 99.90/99.99 | 99.95/100.0 | 99.90/100.0 | 99.78/100.0 | 99.87/100.0 |
| 0.10 | 99.95/100.0 | 99.95/100.0 | 99.95/100.0 | 99.90/99.99 | 99.95/100.0 | 99.90/100.0 | 99.80/100.0 | 99.91/100.0 |

**Effect of different kernel functions on mid-high frequency.** We use the two-degree linear function (quadratic function) as the kernel function $k(f)$ to achieve the enhanced DEFEND representation during the evaluation above. To examine whether this is an optimal function, we conduct further ablation studies by using different kernel functions to fit the frequency distributions. We choose the following functions: simple linear function $k(f) = f$, exponential function, and logarithm function. The parameters of exponential and logarithm functions are set by fit to distributions above ($k(f) = 650 \cdot e^{0.3f} - 650$ for exponential and $k(f) = 0.18 \cdot log(0.25f) + 0.48$). The results are presented in Tab. 6. We observe that different kernel functions lead to different performance, which indicates that a suitable weight function is necessary for the enhanced representation, *e.g.*, the exponential function is not a suitable weight function. We analyze that a suitable and desired function should fit the distributions properly with no overfitting or underfitting. The naive functions also cannot achieve competitive performance, such as $k(f) = f$, which could also be explained by underfitting. The logarithm functions achieve impressive performance, and quadratic functions further improve the results, which indicates that our specifically designed frequency-selective function is a suitable function for restraining the less discriminative bands and enhancing those more discriminative ones.

Table 6: **Ablation study on different kernel functions for mid-high frequencies.** We report the ACC/AP on the GenImage dataset, from which we observe that only a suitable weight function can achieve impressive performance.

| Kernel function | Different Diffusion Models in GenImage | | | | | | | Total |
|---|---|---|---|---|---|---|---|---|
| | ADM | Glide | Midjourney | SD-v1.4 | SD-v1.5 | VQDM | Wukong | Avg. |
| $k(f) = f$ | 99.15/99.93 | 99.50/99.91 | 99.30/99.97 | 98.85/99.80 | 99.20/99.97 | 99.10/99.97 | 98.35/99.87 | 99.06/99.92 |
| $k(f) = a \cdot e^{bf} + c$ | 50.00/54.00 | 50.00/54.00 | 50.00/54.00 | 50.00/54.00 | 50.00/54.00 | 50.00/54.00 | 50.00/52.92 | 50.00/53.85 |
| $k(f) = a \cdot log(bf) + c$ | 99.75/99.99 | 99.95/100.0 | 99.95/100.0 | 99.95/100.0 | 99.90/99.97 | 99.90/99.98 | 99.90/99.98 | 99.90/99.99 |
| $k(f) = af^2 + bf + c$ | 99.95/100.0 | 99.95/100.0 | 99.95/100.0 | 99.90/99.99 | 99.95/100.0 | 99.90/100.0 | 99.80/100.0 | 99.91/100.0 |

## 4.4 VISUALIZATION

To analyze our designed representation more directly, we visualize the Fourier spectrum and our designed representation on real and different diffusion-generated images, as shown in Fig. 6. We observe that our designed representations remove the low-frequency information, which is less discriminative, and that they enhance the high-frequency clues, such as edges and details, which are more discriminative. The representations on real images preserve more mid-high-frequency information of original images compared to diffusion-generated images that are more distinguishable serving as clues for the detection task.

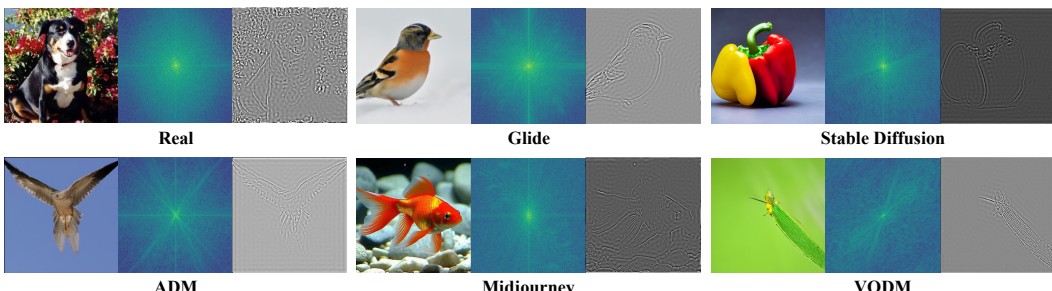

Figure 6: **The visualization of Fourier spectrum and our designed representation** on real and different diffusion-generated images. We observe that our representation enhances the mid-high-frequency clues and removes low-frequency information, which makes it more discriminative for distinguishing real and fake images.

## 5 CONCLUSION

In this paper, we focus on the intrinsic statistical difference between natural real images and diffusion-generated images in the frequency domain. Specifically, we first conduct a comprehensive frequency analysis that shows that the diffusion-generated images exhibit increasing differences with natural real images from low- to high-frequency bands. Upon this observation, we propose a new image representation **DEFEND** by designing a specific frequency-selective function that serves as the weighted filter banks on the Fourier spectrum to restrain the less-discriminative frequency bands, *low-frequency* and to enhance the more discriminative ones, *high-frequency*. Extensive experiments on various public diffusion-generated image datasets demonstrate the superiority of our proposed method with impressive generalization and robustness against other state-of-the-art competitors. We hope our method could provide insights for detecting generated images from the perspective of analyzing natural real images, *i.e.,* in the frequency domain. In the future, we aim to extend our idea and method to other AI-generated content (AIGC) detection tasks, such as different generative models to facilitate the development of AIGC safety.

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
