APPENDIX

In this supplementary material, we provide more details about the preliminaries, related datasets, baselines, additional experimental results, and further ethical discussions about how our method could contribute to the community, described in detail as follows.

The code will be released at https://github.com/anonymous/DEFEND_ICLR_3874.

## A PRELIMINARIES ON DIFFUSION MODELS

Diffusion Models have achieved remarkable success in image synthesis tasks (Ho et al., 2020; Dhariwal & Nichol, 2021; Rombach et al., 2022). The main idea of diffusion models is inspired by the non-equilibrium thermodynamics proposed in (Sohl-Dickstein et al., 2015). Typically, diffusion models define two Markov chains of diffusion steps that first slowly add Gaussian noise to clean images, until disturbing them into isotropic Gaussian noise (termed diffusion or forward process); then they learn to reverse the diffusion process to generate clean samples from the noise (termed denoising or reverse process).

During the diffusion (or forward) process, a clean image sample $\mathbf{x}_0$ is gradually corrupted by adding Gaussian noise in sequential steps $t = 1, \ldots, T$ following a Markov chain process that can be defined as follows:

$$q(\mathbf{x}_t|\mathbf{x}_{t-1}) = \mathcal{N}(\mathbf{x}_t; \sqrt{\frac{\alpha_t}{\alpha_{t-1}}}\mathbf{x}_{t-1}, (1 - \frac{\alpha_t}{\alpha_{t-1}}\mathbf{I})), \tag{12}$$

in which $\mathbf{x}_t$ is the noisy image at $t$-th time step, and $\alpha_t$ is the predefined noise schedule, with $T$ denotes the total steps. We can obtain $\mathbf{x}_t$ from $\mathbf{x}_0$, according to the properties of the Markov process and Gaussian distribution, directly by:

$$q(\mathbf{x}_t|\mathbf{x}_0) = \mathcal{N}(\mathbf{x}_t; \sqrt{\alpha_t}\mathbf{x}_0, (1 - \alpha_t)\mathbf{I})). \tag{13}$$

In the denoising (or reverse) process, the noisy image (or pure Gaussian distribution) is gradually denoised to obtain a clean image that is also a Markov chain and could be defined as:

$$p_\theta(\mathbf{x}_{t-1}|\mathbf{x}_t) = \mathcal{N}(\mathbf{x}_{t-1}; \mu_\theta(\mathbf{x}_t, t), \mathbf{\Sigma}_\theta(\mathbf{x}_t, t)), \tag{14}$$

where the $p_\theta(\mathbf{x}_{t-1}|\mathbf{x}_t)$ is parameterized by neural networks to approximate the underlying data distribution $q(\mathbf{x}_{t-1}|\mathbf{x}_t)$. During training, a neural network $\epsilon_\theta$ is optimized to predict the added noise, given the noisy image $\mathbf{x}_t$ and corresponding time step $t$, the optimization target is a sampling and denoising process that can be defined as follows:

$$L_\theta(\mathbf{x}_0, t) = \|\epsilon - \epsilon_\theta(\sqrt{\bar{\alpha}_t}\mathbf{x}_0 + \sqrt{1 - \bar{\alpha}_t}\epsilon, t)\|^2, \tag{15}$$

where $\epsilon \sim \mathcal{N}(\mathbf{0}, \mathbf{I})$.

## B MORE DETAILS ABOUT THE BASELINES

We present a brief description of state-of-the-art baselines for our comparisons in Sec. 4.1. Here, for better comparison, we summarize more details of each method of detector type, modality, and dependency, as shown in Tab. 7 and described below: 1) ResNet-50 (He et al., 2016) with binary cross-entropy loss is a widely-used backbone for including image classification. 2) Swin-Transformer (Liu et al., 2021), a hierarchical transformer with shifted windows for downstream vision tasks. We choose Swin-B/224×224 as our baseline. 3) Patchforensics (Chai et al., 2020) proposes a patch-wise classifier for detection at patch level. 4) F3Net (Qian et al., 2020) proposes to mine two complementary frequency-aware clues with a two-stream network. 5) DIRE (Wang et al., 2023) introduces a reconstruction error between the original and diffusion-reconstructed image as representation to train the classifier. 6) CNNDet (Wang et al., 2020) carefully designs pre- and post-preprocessing and data augmentation to detect CNN-generated images. We choose Blur+JPEG (0.1) setting as our baseline. 7) UniFD (Ojha et al., 2023) uses CLIP to extract only the image embeddings with the nearest neighbor as the classification head. 8) NPR (Tan et al., 2024) explores the artifacts left by up-sampling layers in GAN and diffusion models to serve as discriminative clues.

Table 7: Details of all baselines and our proposed method.

| Method | Detector Type | Modality | Dependency |
|---|---|---|---|
| ResNet-50 (He et al., 2016) | Backbone | Image | training image |
| Swin-T (Liu et al., 2021) | Backbone | Image | training image |
| Patchfor (Chai et al., 2020) | Deepfakes | Image | specific local patch patterns |
| F3Net (Qian et al., 2020) | Deepfakes | Frequency | specific frequency patterns |
| DIRE (Wang et al., 2023) | Diffusion Models | Image | reconstruction model |
| CNNDet (Wang et al., 2020) | CNN generator | Image | CNN-based generator |
| UniFD (Ojha et al., 2023) | CNN+Diffusion Models | Image | pretrained model & reference set |
| NPR (Tan et al., 2024) | GAN+Diffusion Models | Image | up-sampling operation |
| DEFEND (ours) | Diffusion Models | Frequency | none |

## C  MORE DETAILS ABOUT THE DATASETS

In this section, we describe more details about the training and testing diffusion datasets we used for evaluation. As described in Section. 4.1, to evaluate our method and other baselines, we use totally more than 25 different diffusion models with various settings from (Ojha et al., 2023; Wang et al., 2023; Zhu et al., 2024) . For better comprehension and comparison, we categorize them into different denoising conditions, training image sources, and resolutions, as listed in Tab. 8 (the unconditional/conditional ADM are generated under different settings).

| Dataset | Image Source | Generative Model | Denoising Condition | Resolution |
|---|---|---|---|---|
| GenImage | ImageNet (Russakovsky et al., 2015) | ADM (Dhariwal & Nichol, 2021) | Conditional | 256 × 256 |
| | | Glide (Nichol et al., 2021) | | 256 × 256 |
| | | Midjourney (Midjourney, 2023) | | 1024 × 1024 |
| | | SD-v1.4 (Rombach et al., 2022) | Text-to-Image | 512 × 512 |
| | | SD-v1.5 (Rombach et al., 2022) | | 512 × 512 |
| | | VQDM (Gu et al., 2022) | | 256 × 256 |
| | | Wukong (Wukong, 2022) | | 512 × 512 |
| UniformerDiffusion | ImageNet (Russakovsky et al., 2015) | ADM (Dhariwal & Nichol, 2021) | Conditional | 256 × 256 |
| | | LDM (Rombach et al., 2022) | | 256 × 256 |
| | | Glide (Nichol et al., 2021) | Text-to-Image | 256 × 256 |
| | | DALLE (Ramesh et al., 2021) | | 256 × 256 |
| DiffusionForensics | LSUN (Yu et al., 2015) | ADM (Dhariwal & Nichol, 2021) | Unconditional | 256 × 256 |
| | | DDPM (Ho et al., 2020) | | 256 × 256 |
| | | iDDPM (Nichol & Dhariwal, 2021) | | 256 × 256 |
| | | PNDM (Liu et al., 2022) | | 256 × 256 |
| | | SD-v1 (Rombach et al., 2022) | | 512 × 512 |
| | | SD-v2 (Rombach et al., 2022) | | 768 × 768 |
| | | LDM (Rombach et al., 2022) | | 256 × 256 |
| | | VQDM (Gu et al., 2022) | Text-to-Image | 256 × 256 |
| | | IF (Saharia et al., 2022) | | 256 × 256 |
| | | DALLE2 (Ramesh et al., 2022) | | 1024 × 1024 |
| | | Midjourney (Midjourney, 2023) | | 1024 × 1024 |

Table 8: **Details of the diffusion models for our evaluation** (Wang et al., 2023; Ojha et al., 2023; Zhu et al., 2024), including the training image source, denoising condition, and resolution.

Specifically, our training set includes 40,000 real and 40,000 fake images generated from ADM when trained on ImageNet (Russakovsky et al., 2015); and the test set of most generative model includes 1,000 real and 1,000 fake images (except DALLE2 includes 500 and Midjourney from DiffusionForensics includes 100 fake with an equal number of real images). The resolution of most generated images is 256 × 256 (*e.g.*, ADM, DDPM, PNDM *etc.*). For the images with a higher resolution (*e.g.*, SD-v1, SD-v2, DALLE2, and Midjourney), the generated images are resized into 256 × 256 with bicubic interpolation. Note that the real images are from the corresponding training set of each generative model, unless specifically stated. Moreover, we present examples from each generative model for better comprehension in Fig. 7.

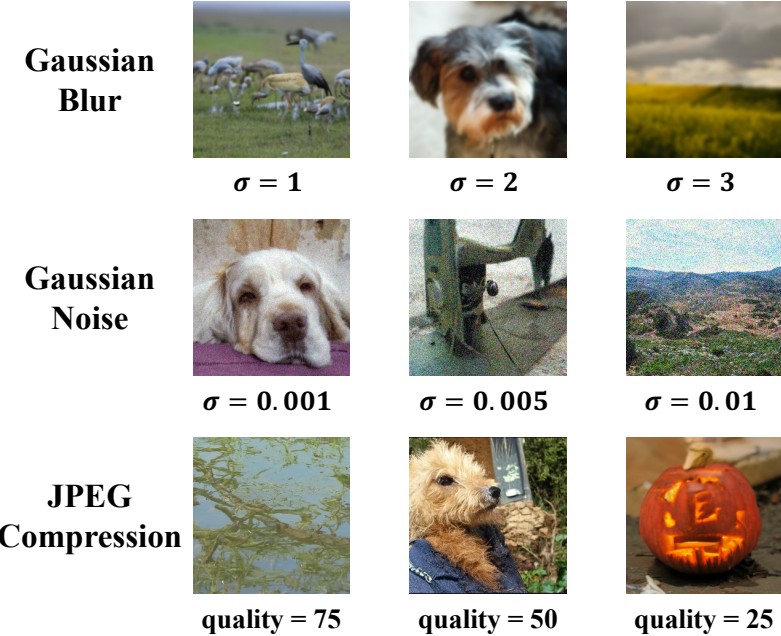

Figure 7: **Examples from different diffusion models**, including GenImage (Zhu et al., 2024), UniformerDiffusion (Ojha et al., 2023), and DiffusionForensics (Wang et al., 2023).

## D EXAMPLES UNDER PERTURBATIONS

In Section. 4.2, we evaluate the robustness of all baselines and our proposed method under three different types of perturbations (Gaussian Noise, Gaussian Blur, and JPEG Compression). For each perturbation, we employ three different severity levels. In this section, we present examples under each perturbation and severity level in Fig. 8 for better comprehension.

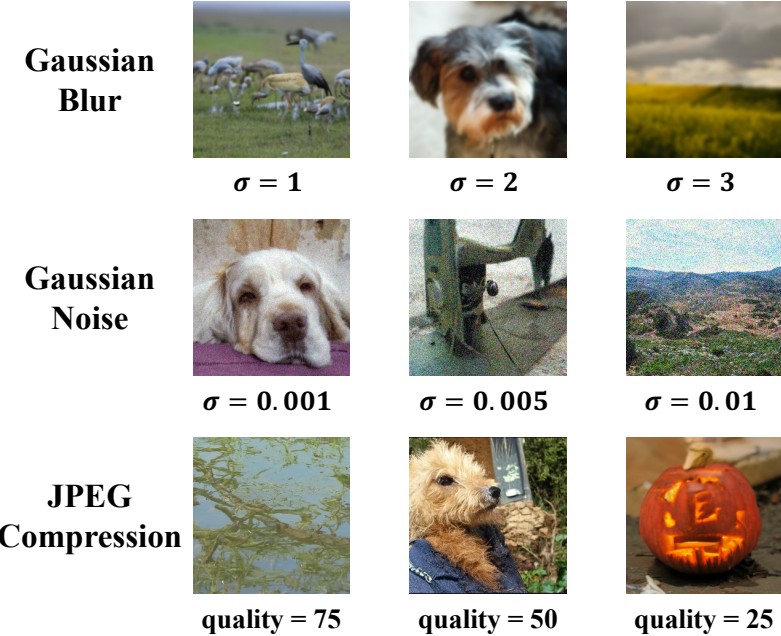

Figure 8: **Examples under three different perturbations** with three different severity levels, including Gaussian Blur, Gaussian Noise, and JPEG compression.

## E ADDITIONAL EXPERIMENTS

### E.1 EFFECT OF DIFFERENT LINEAR DEGREES FOR KERNEL FUNCTION

To evaluate the effect of linear function on performance, we conduct ablation experiments by employing linear function with different degrees, *i.e.*, linear/quadratic/cubic functions. The results are shown in Tab. 9 (the coefficients for three- and one-degree linear functions are: $k(f) = -4f^3 +$

$3.2f^2 - 0.16f + 0.02$ and $k(f) = f - 0.04$), from which we observe that both low- and high-degree linear functions lead to slight performance drops. One explanation could be that both the too simple and the complicated linear functions cannot fit the distributions properly, *i.e.*, with underfitting for the low degree functions and overfitting for the high-degree ones. The comparisons also demonstrate that, to obtain the desired representation, a two- degree linear function is a suitable choice for restraining and enhancing different frequency bands. Moreover, we also believe that if other functions can fit the distribution properly, they can also be employed to obtain the enhanced representation and achieve competitive performance.

Table 9: **Ablation study on different linear degrees.** We report the ACC/AP results on the GenImage dataset, from which we observe that both too simple and complex linear functions can lead to a slight performance drop.

| Kernel function | Different Diffusion Models in GenImage | | | | | | | Total |
|---|---|---|---|---|---|---|---|---|
| | ADM | Glide | Midjourney | SD-v1.4 | SD-v1.5 | VQDM | Wukong | Avg. |
| $k(f) = af^3 + bf^2 + cf + d$ | 100.0/100.0 | 99.90/100.0 | 99.95/100.0 | 99.90/99.99 | 99.80/99.99 | 99.85/100.0 | 99.40/99.99 | 99.83/100.0 |
| $k(f) = af + b$ | 99.85/100.0 | 99.90/100.0 | 99.80/99.99 | 99.50/99.96 | 99.70/99.99 | 99.70/100.0 | 99.10/99.98 | 99.65/99.99 |
| $k(f) = af^2 + bf + c$ | 99.95/100.0 | 99.95/100.0 | 99.95/100.0 | 99.90/99.99 | 99.95/100.0 | 99.90/100.0 | 99.80/100.0 | 99.91/100.0 |

# F    ETHICAL DISCUSSIONS

With the development of current generative models, the competition between generation and detection is always in progress. Prior diffusion detectors might suffer from the upcoming new diffusion models, and the new diffusion models can promote the development of new detectors. Our method is based on the general observation and analysis of the frequency difference between natural real images and diffusion-generated images. To achieve general and robust detection, we further propose to enhance the discriminative frequency bands and restrain the less discriminative ones. If the diffusion models in the future completely improve or change the noising/denoising process, which could cause entirely different frequency distributions of diffusion-generated images and should be difficult to achieve, all methods based on frequency traces or other current diffusion-related characteristics might fail. Nevertheless, we believe our method can still provide insight into the general and robust detection of diffusion-generated images from the perspective of natural real image distribution.