# OpenReview forum: "Leveraging Natural Frequency Deviation for Diffusion-Generated Image Detection"
_ICLR.cc/2025/Conference — ICLR 2025 Conference Withdrawn Submission_

### Official Review · Reviewer_fjNC · 2024-10-21

**Soundness:** 1
**Presentation:** 1
**Contribution:** 1
**Rating:** 3
**Confidence:** 4

**Summary:**

This paper studies the recent problem of detecting images generated by diffusion models. The base idea is to utilize the representation of the frequency domain. For better representation, the authors propose to manually weight the frequency components via pre-designed weight mapping and use the inverse DFT output for the final map for training. The proposed method shows near-perfect results in all benchmarks.

**Strengths:**

1) Detecting data generated from generative models from real data has accumulated great societal interest.

2) If the results are true, it will nearly solve the deepfake detection problem in existing benchmarks.

**Weaknesses:**

1. The main idea of the paper is not novel. Overall, utilizing the frequency domain for deepfake detection (especially in the high-frequency component) has been exhaustively explored. Although the authors put a lot of effort into explaining the power spectrum and the design, the final form of Equation (4) shares similarities with previous papers that utilize frequency domain representation (to cite a few, [1~4]). Furthermore, the idea of applying the power spectrum analysis for deepfake detection has been introduced in previous literature ([5]).

2. Connected to 1., the paper proposes a weighted filter as a main difference compared to existing frequency-based methods. The main difference is the pre-defined quadratic weight and threshold in Equation (10), which seems to be set "Empirically". I do not find any meaningful justification for the choice. Hence, no justification for the benefit of the proposed method is introduced, and it is hard to understand why this method works.

3. However, all of a sudden, the result in this paper is near-perfect. It is hard to understand why introducing quadratic frequency weighting term nearly solves the challenging benchmarks while previous methods that engineer the weighting function by learning (e.g. [1]) do not. Given that there is no code available, the results in this paper do not satisfy my concerns.

4. I also find the experiment setting is distant from common practices. For example, the GenImage benchmark usually trains on the ImageNet training dataset and SDv1.4-generated images. This paper trains on ADM images for AI-generated images. It is hard to compare the results reported in this paper to the other papers, which further hinders resolving my concerns about the experiment results.

5) While this seems relatively minor, I do not concur with the author's proposition in related works in L130-132. I think the existing works also take account of the distribution of the real images since they train their classifiers on the real data's frequency representation.


Overall, I feel the idea of the paper is dated and the methodology shares similarities to various existing works. The only difference is the quadratic weight, and I do not understand why this difference should critically work. Given that, I find it hard to understand why the reported results show near-perfect performance.

**Questions:**

1. Can you report the results of [1] and [3] in the GenImage dataset as a baseline? I think the proposed methods apply more sophisticated mapping, and they should perform similarly to the DEFEND.

2. How did the authors select the quadratic term empirically?

3. What's the performance in the GenImage dataset when SDv1.4 is used for training?

4. Why the BigGAN dataset is excluded? Analysis on why the method can or cannot generalize on the "GAN-based methods" would be helpful to further understand the method.


---- References ----

[1] Frequency-Aware Deepfake Detection: Improving Generalizability through
Frequency Space Domain Learning, Tan et al, 2024

[2] FrePGAN: Robust Deepfake Detection Using Frequency-Level Perturbations, Jeong et al., 2022


[3] Dynamic Graph Learning with Content-guided Spatial-Frequency Relation
Reasoning for Deepfake Detection, Wang et al., 2023


[4] A SANITY CHECK FOR AI-GENERATED IMAGE DETECTION, Yan et al., 2024
[5] Unmasking DeepFakes with simple Features, Durall et al., 2019.

---

### Official Review · Reviewer_cDMB · 2024-10-31

**Soundness:** 2
**Presentation:** 2
**Contribution:** 2
**Rating:** 3
**Confidence:** 4

**Summary:**

This paper first conducts a comprehensive frequency analysis on natural real images and diffusion-generated fake images.
Then the authors design a representation called DEFEND, which is a frequency-selective function that serves as weighted filter banks for restraining the less discriminative bands (i.e., low-frequency) and enhancing the more discriminative frequency bands (i.e., high-frequency) in the Fourier spectrum.
Experimental results show that DEFEND outperforms other state-of-the-art methods in detecting diffusion-generated images.

**Strengths:**

1. The paper is well-written and easy to follow.
2. Using frequency analysis to detect diffusion-generated images is reasonable.
3. The experiments are done to demonstrate the effectiveness of the proposed method.

**Weaknesses:**

1. The proposed method is not very innovative. The idea of using frequency analysis to detect deepfakes is not new.

Zhang J, Wang Y, Tohidypour H R, et al. Detecting Stable Diffusion Generated Images Using Frequency Artifacts: A Case Study on Disney-Style Art[C]//2023 IEEE International Conference on Image Processing (ICIP). IEEE, 2023: 1845-1849.

Ricker J, Damm S, Holz T, et al. Towards the detection of diffusion model deepfakes[J]. arXiv preprint arXiv:2210.14571, 2022.

Bammey Q. Synthbuster: Towards detection of diffusion model generated images[J]. IEEE Open Journal of Signal Processing, 2023.

2. The paper is not very appealing and lacks sufficient insight.

3.The evaluation benchmark may be too simple and not very convincing. Many methods achieve 99\% mAP, which may not be enough to show the superiority of the method.

**Questions:**

See Weaknesses

---

### Official Review · Reviewer_M8Kb · 2024-11-03

**Soundness:** 4
**Presentation:** 3
**Contribution:** 3
**Rating:** 6
**Confidence:** 3

**Summary:**

The paper addresses the challenge of detecting images generated by diffusion models. The authors propose a novel image representation called Natural Frequency Deviation (DEFEND). This method is based on the observation that diffusion-generated images show progressively larger differences from real images across low to high-frequency bands in the Fourier spectrum. DEFEND applies a weighted filter to the Fourier spectrum to suppress less discriminative bands (low frequency) and enhance more informative ones (high frequency). Extensive experiments on various benchmark datasets demonstrate the effectiveness of the proposed method.

**Strengths:**

1. The paper introduces a new representation (DEFEND) that leverages frequency domain analysis to distinguish between real and diffusion-generated images,  which is designed to be general and robust, capable of detecting images from unseen diffusion models.

2. The paper provides a thorough frequency analysis comparing real and diffusion-generated images, which adds depth to the understanding of the problem.

3. Extensive experiments on multiple datasets demonstrate the superiority of DEFEND over state-of-the-art detectors.

**Weaknesses:**

1. The DEFEND method might be more complex to implement compared to simpler detection methods, which could be a barrier for some users.

2.  The paper relies heavily on the assumption that low-frequency bands are less discriminative and high-frequency bands are more discriminative. This assumption may not hold in all scenarios or for all types of images.

**Questions:**

1 In Figure 2c, 1000 steps of denoising are performed, but only 900 steps are drawn. What will happen if the number of steps is lower?

2 The spectrum of an image is also related to the content of the image. Did the article consider the performance of different image contents when analyzing? In the analysis method in Figure 1, what are the differences in the spectrum of different contents of the same category of images (real or fake)?

The author can consider adding an experiment where the fake image is constructed using a image2image diffusion model, such as inputting a natural image into a diffusion-based model to generate a image with a similar context, and then distinguishing it, thereby eliminating the influence of the class content in the image. ,

---

### Official Review · Reviewer_SMqS · 2024-11-04

**Soundness:** 3
**Presentation:** 3
**Contribution:** 3
**Rating:** 6
**Confidence:** 3

**Summary:**

This paper proposes a synthetic image detection method with frequency analysis, based on the discrepancy of the high-frequency components of real and synthetic images. According to the motivating power spectrum analysis, the synthetic images (generated with diffusion models) under-represent high frequency components compared to real images, especially in low-noise regime (last steps in diffusion generation). The paper points out this phenomenon, and this is caused in the same manner that the likelihood-maximizing loss function for diffusion model do not work prominently for high-fidelity image synthesis.
The detection method is as follows: first, after sampling from the pre-trained diffusion model, the paper first measures the discrete Fourier transform (DFT) of gray-scaled natural/diffusion-generated images then compare the mean discrepancy in terms of power spectrum in the dataset. Then with the designed weight (focuses in high frequency) multiplicated with the frequency components is considered as the classifier (detector) input. In various image datasets, the proposed method achieves almost perfect performance, while existing works does not.

**Strengths:**

* The insight makes sense: The Gaussian distribution has the same amplitude across all frequency bands, while the image dataset has decaying power spectrum with higher frequency bands. As the diffusion model that predicts the (expected) noises
* If the performance reported in this paper is correct, this paper __finalizes__ the diffusion image detection task, with 99.99-100.00% accuracy and average precision for all datasets.

**Weaknesses:**

See questions.

**Questions:**

* The legends in Figure 2(b) and 4(b) are duplicating.
* (Minor) is the title "Exploiting" or "Leveraging"?
* I have compared the paper's main result to [Ojha, 2023; Tan, 2024], and the numbers of the experimental results in Table 2 (DiffusionForensic) is relatively higher than that of Table 3 in [Tan, 2024]. What is the difference in the experimental setting of this paper, compared to those prior works?
* Can we see the error bar of the frequency components in Figure 2?
* In (Line 186-192), the paper pointed out that the underestimation of the end steps is caused by the "weight-free" optimization objective described in Equation (2). Then, what happens if the likelihood-maximizing objective is used instead of the widely-used cases that the weights w.r.t. time is equal (like in Eq (2))?

```
[Ojha, 2023] Towards Universal Fake Image Detectors that Generalize Across Generative Models, CVPR 2023
[Tan, 2024] Rethinking the up-sampling operations in cnn-based generative network for generalizable deepfake detection, CVPR 2024
```

---

### Note · Authors · 2024-11-15

I have read and agree with the venue's withdrawal policy on behalf of myself and my co-authors.